# Fast Convergence of Langevin Dynamics on Manifold: Geodesics meet Log-Sobolev

**Xiao Wang**
SUTD
xiao_wang@sutd.edu.sg

**Qi Lei**
Princeton
qilei@princeton.edu

**Ioannis Panageas**
UC Irvine
ipanagea@uci.edu

## Abstract

Sampling is a fundamental and arguably very important task with numerous applications in Machine Learning. One approach to sample from a high dimensional distribution $e^{-f}$ for some function $f$ is the Langevin Algorithm (LA). Recently, there has been a lot of progress in showing fast convergence of LA even in cases where $f$ is non-convex, notably Vempala and Wibisono [2019], Moitra and Risteski [2020] in which the former paper focuses on functions $f$ defined in $\mathbb{R}^n$ and the latter paper focuses on functions with symmetries (like matrix completion type objectives) with manifold structure. Our work generalizes the results of Vempala and Wibisono [2019] where $f$ is defined on a manifold $M$ rather than $\mathbb{R}^n$. From technical point of view, we show that KL decreases in a geometric rate whenever the distribution $e^{-f}$ satisfies a log-Sobolev inequality on $M$.

## 1 Introduction

We focus on the problem of sampling from a distribution $e^{-f(x)}$ supported on a Riemannian manifold $M$ with standard volume measure. Sampling is a fundamental and arguably very important task with numerous applications in machine learning and Langevin dynamics is a quite standard approach. There is a growing interest in Langevin algorithms, e.g. Welling and Teh [2011], Wibisono [2018], Dalalyan [2017a], due to its simple structure and the good empirical behavior. The classic Riemannian Langevin algorithm, e.g. Girolami and Calderhead [2011], Patterson and Teh [2013], Zhang et al. [2020], is used to sample from distributions supported on $\mathbb{R}^n$ (or a subset $D$) by endowing $\mathbb{R}^n$ (or $D$) a Riemannian structure. Beyond the classic application of Riemannian Langevin Algorithm (RLA), recent progress in Domingo-Enrich et al. [2020], Moitra and Risteski [2020], Li and Erdogdu [2020] shows that sampling from a distribution on a manifold has application in matrix factorization, principal component analysis, matrix completion, solving SDP, mean field and continuous games and GANs. Formally, a game with finite number of agents is called continuous if the strategy spaces are continuous, either a finite dimensional differential manifold or an infinite dimensional Banach manifold Ratliff et al. [2013, 2016], Domingo-Enrich et al. [2020]. The mixed strategy is then a probability distribution on the strategy manifold and mixed Nash equilibria can be approximated by Langevin dynamics.

**Geodesic Langevin Algorithm (GLA).** In order to sample from a distribution on $M$, geodesic based algorithms (e.g. Geodesic Monte Carlo and Geodesic MCMC) are considered in Byrne and Girolami [2013], Liu et al. [2016], where a geodesic integrator is used in the implementation. We propose a Geodesic Langevin Algorithm (GLA) as a natural generalization of unadjusted Langevin algorithm (ULA) from the Euclidean space to manifold $M$. The benefit of GLA is to leverage sufficiently the geometric information (curvature, geodesic distance, isoperimetry) of $M$ while keeping the structure of the algorithm simple enough, so that we can obtain a non-asymptotic convergence guarantee of the algorithm. In local coordinate systems, the Riemannian metric is represented by a matrix $g = \{g_{ij}\}$, see Definition 3.3. We denote $g^{ij}$ the $ij$-th entry of the inverse

matrix $g^{-1}$ of $g$, and $|g| = \det(g_{ij})$, the determinant of the matrix $\{g_{ij}\}$. Then GLA is the stochastic process on $M$ that is defined by

$$x_{k+1} = \operatorname{Exp}_{x_k}(\epsilon F + \sqrt{2\epsilon g^{-1}}\xi_0) \tag{1}$$

where $F = (F_1, ..., F_n)$ with

$$F_i = -\sum_j g^{ij}\frac{\partial f}{\partial x_j} + \frac{1}{\sqrt{|g|}}\sum_j \frac{\partial}{\partial x_j}\left(\sqrt{|g|}g^{ij}\right), \tag{2}$$

$\epsilon > 0$ is the stepsize, $\xi_0 \sim \mathcal{N}(0, I)$ is the standard Gaussian noise, and $\operatorname{Exp}_x(\cdot) : T_x M \to M$ is the exponential map (Definition 3.5). Clearly GLA is a two-step discretization scheme of the Riemannian Langevin equation

$$dX_t = F(X_t)dt + \sqrt{2g^{-1}}dB_t$$

where $F$ is given by (2). Suppose the position at time $k$ is $x_k$, then the next position $x_{k+1}$ can be obtained by the following tangent-geodesic composition:

1. Tangent step: Take a local coordinate chart $\varphi : U_{x_k} \to \mathbb{R}^n$ at $x_k$, this map induces the expression of $g_{ij}$ and $g^{ij}$, then compute the vector $v = \epsilon F + \sqrt{2\epsilon g^{-1}}\xi_0$ in tangent space $T_{x_k}M$;

2. Geodesic step: Solve the geodesic equation (a second order ODE) whose solution is a curve $\gamma(t) \subset \varphi(U_{x_k})$, such that the initial conditions satisfy $\gamma(0) = \varphi(x_k)$ and $\gamma'(0) = v$. Then let $x_{k+1} = \gamma(1)$ be the updated point.

The exponential map and ODE solver for geodesic equations is commonly used in sampling algorithms on manifold, e.g. Vempala and Lee [2017], Byrne and Girolami [2013], Liu et al. [2016]. We will discuss on other approximations of the exponential map without solving ODEs through illustrations in a later section. Figure 1 gives an intuition of GLA on the unit sphere where the exponential map is $\operatorname{Exp}_x(v) = \cos(\|v\|)x + \sin(\|v\|)\frac{v}{\|v\|}$.

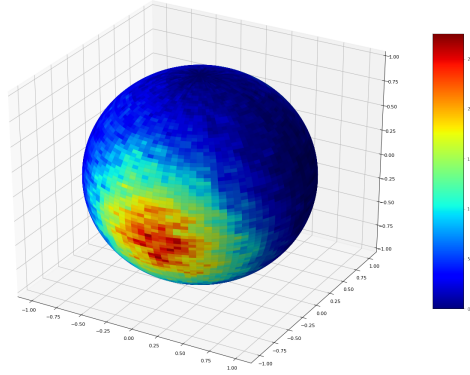

Figure 1: $f(x_1, x_2, x_3) = x_1^2 + 3.05x_2^2 - 0.9x_3^2 + 1.1x_1x_2 + -1.02x_2x_3 + 2.1x_3x_1$, $\epsilon = 0.1$, Iterations: 100k.

The main result on convergence is stated as follows.

**Theorem 1.1** (Informal). *Let $M$ be a closed $n$-dimensional manifold (Definition 3.2). Suppose that $\nu = e^{-f(x)}$ is a distribution on $M$ with $\alpha > 0$ the log-Sobolev constant. Then there exists a real number $K_2, K_3, K_4, C$, such that by choosing stepsize $\epsilon$ properly based on the Lipschitz constant of the Riemannian gradient of $f$, log-Sobolev constant of the target distribution $\nu$, dimension and curvature of $M$, the KL divergence $H(\rho_k|\nu)$ decreases along the GLA iterations rapidly in the sense that*

$$H(p_k|\nu) \le e^{-\alpha k\epsilon}H(p_0|\nu) + \frac{16\epsilon}{3\alpha}(2nL^2 + 2n^3K_2C + nK_3K_4).$$

The same as unadjusted Langevin algorithm (ULA) in Euclidean space, GLA is a biased algorithm that converges to a distribution different from $e^{-f(x)}$. Practically we need a lower bound estimate for $\alpha$. With additional condition on Ricci curvature, this lower bound can be chosen based on the diameter of $M$ by Theorem **??**.

Our main technical contributions are:

- A non-asymptotic convergence guarantee for Geodesic Langevin algorithm on closed manifold is provided, with the help of log-Sobolev inequality.

- The framework of this paper serves as the first step understanding to the rate of convergence in sampling from distributions on manifold with log-Sobolev inequality, and can be generalized to prove non-asymptotic convergence results for more general settings and more subtle algorithms, i.e., for open manifolds and unbiased algorithms.

**Comparison to literarture** The typical difference between algorithm (1) and the classic RLA is the use of exponential map. As $\epsilon \to 0$, both GLA and RLA boil down to the same continuous time Langevin equation in the local coordinate system:

$$dX_t = F(X_t)dt + \sqrt{2g^{-1}}dB_t$$

where $F$ is given by (2) and $B_t$ is the standard Brownian motion in $\mathbb{R}^n$. The direct Euler-Maryuyama discretization iterates in the way that $x_{k+1} = x_t + \epsilon F(x_k) + \sqrt{2\epsilon g^{-1}(x_k)}\xi_0$. However, by adding the vector $\epsilon F(x_k) + \sqrt{2\epsilon g^{-1}(x_k)}\xi_0$ that is in the tangent space to a point $x_k$ that is on the manifold $M$ has no intrinsic geometric meaning, since the resulted point $x_{k+1}$ is indeed not in $M$. The exponential map just gives a way to pull $x_{k+1}$ back to $M$. On the other hand, since RLA is firstly used to sample from distributions on $\mathbb{R}^n$ (or its domain) with a Riemannian structure, Roberts and Stramer [2002], Girolami and Calderhead [2011], Patterson and Teh [2013], Smith et al. [2018], this requires a global coordinate system of $M$, i.e. $M$ is covered by a single coordinate chart and the iterations do not transit between different charts. But this makes it difficult to use RLA when there are inevitably multiple coordinate charts on $M$. More sophisticated algorithms like Geodesic MCMC Liu et al. [2016] is used to transit between different coordinate charts, but to the best knowledge of the authors, the rate of convergence is missing in the literature. Li and Erdogdu Li and Erdogdu [2020] generalize the result of Vempala and Wibisono [2019] by implementing the Riemannian Langevin algorithm in two steps (gradient+Riemannian Brownian motion).

## 2 Related Works

Unadjusted Langevin algorithm (ULA) when sampling from a strongly logconcave density in Euclidean space has been studied extensively in the literature. The bounds for ULA is known in Cheng and Bartlett [2018], Dalalyan [2017a], Dalalyan and Karagulyan [2019], Durmus et al. [2018]. The case when $f$ is strongly convex and has Lipschitz gradient is studied by Dalalyan [2017b], Durmus and Moulines [2017], Durmus and Mounline [2019]. Since ULA is biased because of the discretization, i.e. it converges to a limit distribution that is different from that from continuous Langevin equation. the Metropolis-Hastings correction is widely used to correct this bias, e.g. Roberts and Tweedie [1996], Dwivedi et al. [2018]. A simplified correction algorithm is proposed by Wibisono [2018] that is called symmetrized Langevin algorithm with a smaller bias than ULA. Convergence results is obtained for Proximal Langevin algorithm (PLA) in Wibisono [2019]. In the case where the target distribution is log-concave, there are other algorithms proven to converge rapidly, i.e., Langevin Monte Carlo by Bernton [2018], ball walk and hit-and-run Kannan et al. [1997], Lovász and Vempala [2006a, 2007, 2006b], and Hamiltonian Monte Carlo by Durmus et al. [2017], Vempala and Lee [2018], Mangoubi and Vishnoi [2018]. The underdamped version of the Langevin dynamics under log-Sobolev inequality is studied by Ma et al. [2019], where an iteration complexity for the discrete time algorithm that has better dependence on the dimension is provided. A coupling approach is used by Eberle et al. [2018] to quantify convergence to equilibrium for Langevin dynamics that yields contractions in a particular Wasserstein distance and provides precise bounds for convergence to equilibrium. The case where the densities that are neither smooth nor log-concave is studied in Luu et al. [2017] and asymptotic consistency guarantees is provided. For the Wasserstein distance, Cheng et al. [2018], Majka et al. [2018], Raginsky et al. [2017] provide convergence bound. An earlier research on stochastic gradient Langevin dynamics with application in Bayesian learning is proposed

by Welling and Teh [2011], The Langevin Monte Carlo with a weaker smoothness assumption is studied by Chatterji et al. [2019]. In order to improve sample quality, Gorham and Mackey [2017] develops a theory of weak convergence for kernel Stein discrepancy based on Stein's method. In general, sampling from non log-concave densities is hard, Ge et al. [2018] gives an exponential lower bound on the number of queries required.

The Riemannian Langevin algorithm has been studied in different extent. Related to volume computation of a convex body in Euclidean space, one can endow the interior of a convex body the structure of a Hessian manifold and run geodesic (with respect to the Hessian metric) random walk Vempala and Lee [2017] that is a discretization scheme of a stochastic process with uniform measure as the stationary distribution. The rigorous proof of the convergence of Riemannian Hamiltonian Monte Carlo for sampling Gibbs distribution and uniform distribution in a polytope is given by Vempala and Lee [2018]. In sampling non-uniform distribution, Zhang et al. [2020] gives a discretization scheme related to mirror descent and a non-asymptotic upper bound on the sampling error of the Riemannian Langevin Monte Carlo algorithm in Hessian manifold. The mirrored Langevin is firstly considered by Hsieh et al. [2018] and a non-asymptotic rate is obtained and generalized to the case when only stochastic gradients (mini-batch) are available. An affine invariant perspective of continuous time Langevin dynamics for Bayesian inference is studied in Inigo et al. [2019]. Positive curvature is used to show concentration results for Hamiltonian Monte Carlo in Seiler et al. [2014]. Liu et al. [2019] understand MCMC as gradient flows on Wasserstein spaces and HMC on implicitly defined manifolds is studied in Brubaker et al. [2012].

## 3 Preliminaries

For a complete introduction to Riemannian manifold and stochastic analysis on manifold, we recommend Lee [2018] and Hsu [2002] for references.

### 3.1 Riemannian geometry

**Definition 3.1** (Manifold). A $C^k$-differentiable, $n$-dimensional manifold is a topological space $M$, together with a collection of coordinate charts $\{(U_\alpha, \varphi_\alpha)\}$, where each $\varphi_\alpha$ is a $C^k$-diffeomorphism from an open subset $U_\alpha \subset M$ to $\mathbb{R}^n$. The charts are compatible in the sense that, whenever $U_\alpha \cap U_\beta \neq \emptyset$, the transition map $\varphi_\alpha \circ \varphi_\beta^{-1}(U_\beta \cap U_\alpha) \to \mathbb{R}^n$ is of $C^k$.

**Definition 3.2** (Closed manifold). A manifold $M$ is called *closed* if $M$ is compact and has no boundary.

Typical examples of closed manifolds include sphere and torus.

**Definition 3.3** (Riemannian metric). A Riemannian manifold $(M, g)$ is a differentiable manifold $M$ with a Riemannian metric $g$ defined as the inner product on the tangent space $T_x M$ for each point $x$, $g(\cdot, \cdot) : T_x M \times T_x M \to \mathbb{R}$. Then length of a smooth path $\gamma : [0, 1] \to M$ is $|\gamma| = \int_0^1 \sqrt{g(\gamma'(t), \gamma'(t))} dt$. In a local coordinate chart, $g$ is represented by a $n \times n$ symmetric positive definite matrix with entries $g_{ij}$.

**Definition 3.4** (Geodesic). We call a curve $\gamma(t) : [0, 1] \to M$ a geodesic if it satisfies both of the following conditions:

1. The curve $\gamma(t)$ is parametrized with constant speed, i.e. $\left\| \frac{d}{dt} \gamma(t) \right\|_{\gamma(t)}$ is constant for $t \in [0, 1]$.

2. The curve is the locally shortest length curve between $\gamma(0)$ and $\gamma(1)$, i.e. for any family of curve $c(t, s)$ with $c(t, 0) = \gamma(t)$ and $c(0, s) = \gamma(0)$ and $c(1, s) = \gamma(1)$, we have $\frac{d}{ds}|_{s=0} \int_0^1 \left\| \frac{d}{dt} c(t, s) \right\|_{c(t,s)} dt = 0$.

We use $\gamma_{x \to y}$ to denote the geodesic from $x$ to $y$ ($\gamma_{x \to y}(0) = x$ and $\gamma_{x \to y}(1) = y$). The most important property of a geodesic $\gamma(t)$ is that the time derivative $\dot{\gamma}(t)$ as a vector field, has 0 covariant derivative, i.e. $\nabla_{\dot{\gamma}(t)} \dot{\gamma}(t) = 0$. This property boils down to a second order ODE in local coordinate systems,

$$\ddot{\gamma}_i(t) + \sum_{j,k} \Gamma^i_{jk}(\gamma(t)) \dot{\gamma}_j(t) \dot{\gamma}_k(t) = 0$$

for $i \in [n]$, where $\Gamma^i_{jk}$ are the Christoffel symbols. Given a initial position $\gamma(0)$ and initial velocity $\dot{\gamma}(0)$, by the fundamental theorem of ODE, there exists a unique solution satisfying the geodesic equation. This is the principle we can use the ODE solver in GLA.

**Definition 3.5** (Exponential map). The exponential map $\mathrm{Exp}_x(v)$ is maps $v \in T_x M$ to $y \in M$ such that there exists a geodesic $\gamma$ with $\gamma(0) = x$, $\gamma(1) = y$ and $\gamma'(0) = v$.

The exponential map can be thought of moving a point along a vector in manifold in the sense that the exponential map in $\mathbb{R}^n$ is nothing but $\mathrm{Exp}_x(v) = x + v$. The exponential map on sphere at $x$ with direction $v$ is $\mathrm{Exp}_x(v) = \cos(\|v\|)x + \sin(\|v\|)\frac{v}{\|v\|}$.

**Definition 3.6** (Parallel transport). The parallel transport $\Gamma^y_x$ is a map that transport $v \in T_x M$ to $\Gamma^y_x \in T_x M$ along $\gamma_{x \to y}$ such that the vector stays constant by satisfying a zero-acceleration condition.

Next, we refer the definition of Riemannian gradient and divergence only in local coordinate systems that is used in this paper.

**Definition 3.7** (Gradient and Divergence). In local coordinate system, the gradient of $f$ and the divergence of a vector field $V = \sum_i V_i \frac{\partial}{\partial x_i}$ on a Riemannian manifold is given by

$$\mathrm{grad} f = \sum_{i,j} g^{ij} \frac{\partial f}{\partial x_i} \frac{\partial}{\partial x_i} \quad \text{and} \quad \mathrm{div} V = \frac{1}{\sqrt{|g|}} \sum_i \frac{\partial}{\partial x_i} \left( \sqrt{|g|} V_i \right)$$

where $g^{ij}$ is the $ij$-th entry of the inverse matrix $g^{-1}$ of $g$, $|g| = \det(g_{ij})$.

**Definition 3.8** (Lipschitz gradient). $f$ is of Lipschitz gradient if there exists a constant $L > 0$ such that
$$\|\mathrm{grad} f(y) - \Gamma^y_x \mathrm{grad} f(x)\| \leq L d(x, y) \quad \text{for all} \quad x, y \in M$$
where $d(x, y)$ is the geodesic distance between $x$ and $y$, and $\Gamma^y_x$ is the parallel transport from $x$ to $y$, see Definition 3.6.

## 3.2 Stochastic differential equations

Let $\{X_t\}_{t \geq 0}$ be a stochastic process in $\mathbb{R}^n$ and $B_t$ be the standard Brownian motion in $\mathbb{R}^n$.

**Fokker-Planck Equation** For any stochastic differential equation of the form
$$dX_t = F(X_t, t)dt + \sigma(X_t, t)dB_t,$$
the probability density of the SDE is given by the PDE

$$\frac{\partial \rho(x, t)}{\partial t} = -\sum_{i=1}^n \frac{\partial}{\partial x_i} \left( F_i(x, t)\rho(x, t) \right) + \sum_{i=1}^n \sum_{j=1}^n \frac{\partial^2}{\partial x_i \partial x_j} \left( A_{ij}(x, t)\rho(x, t) \right)$$

where $A = \frac{1}{2}\sigma\sigma^\top$, i.e. $A_{ij} = \frac{1}{2} \sum_{k=1}^n \sigma_{ik}(x, t)\sigma_{jk}(x, t)$

## 3.3 Distributions on manifold

Let $\rho$ and $\nu$ be probability distribution on $M$ that is absolutely continuous with respect to the Riemannian volume measure (denoted by $dx$) on $M$.

**Definition 3.9** (KL divergence). The Kullback-Leibler (KL) divergence of $\rho$ with respect to $\nu$ is

$$H(\rho|\nu) = \int_M \rho(x) \log \frac{\rho(x)}{\nu(x)} dx.$$

**Definition 3.10** (Wasserstein distance). The Wasserstein distance between $\mu$ and $\nu$ is defined to be

$$W_2(\mu, \nu) = \inf\{\sqrt{\mathbb{E}[d(X, Y)^2]} : \mathrm{law}(X) = \mu, \mathrm{law}(Y) = \nu\}.$$

**Definition 3.11** (Talagrand inequality). The probability measure $\nu$ satisfies a Talagrand inequality with constant $\alpha > 0$ if for all probability measure $\rho$, absolutely continuous with respect to $\nu$, with finite moments of order 2,

$$W_2(\rho, \nu)^2 \leq \frac{2}{\alpha} H(\rho|\nu)$$

**Definition 3.12** (Log-Sobolev inequality). A probability measure $\nu$ on $M$ is called to satisfy the logarithmic Sobolev inequality (LSI) if there exists a constant $\alpha > 0$ such that

$$\int_M g^2 \log g^2 d\nu - \left(\int_M g^2 d\nu\right) \log\left(\int_M g^2 d\nu\right) \leq \frac{2}{\alpha} \int_M \|\mathrm{grad} g\|^2 d\nu,$$

for all smooth functions $g : M \to \mathbb{R}$ with $\int_M g^2 \leq \infty$. The largest possible constant $\alpha$ is called the logarithmic Sobolev constant (LSC).

## 4 Main Results

### 4.1 Technical Overview

**Wasserstein gradient flow.** The equivalence between Langevin dynamics and optimization in the space of densities is based on the result of Jordan et al. [1998], Wibisono [2018] that the Langevin dynamics captures the gradient flow of the relative entropy functional in the space of densities with the Wasserstein metric. As a result, running the Langevin dynamics is equivalent to sampling from the stationary distribution of the Wasserstein gradient flow asymptotically. To minimize $\int_M f(x)\rho(x)dx$ with respect to $\rho \in \mathcal{P}(M)$, we consider the entropy regularized functional of $\rho$ defined as follows,

$$\mathcal{L}(\rho) = \mathcal{F}(\rho) + H(\rho)$$

where $\mathcal{F}(\rho) = \int_M f(x)\rho(x)dx$ and $H(\rho) = \int_M \rho(x)\log\rho(x)dx$ that is the negative Shannon entropy $h(\rho) = -\int_M \rho(x)\log\rho(x)dx$. According to Santanmbrogio [2015], the Wasserstein gradient flow associated with functional $\mathcal{L}$ is the Fokker-Planck equation

$$\frac{\partial \rho(x,t)}{\partial t} = \mathrm{div}\left(\rho(x,t)\mathrm{grad}f(x) + \mathrm{grad}\rho(x,t)\right) = \mathrm{div}\left(\rho(x,t)\mathrm{grad}f(x)\right) + \Delta_M \rho(x,t), \quad (3)$$

where $\mathrm{grad}$ and $\mathrm{div}$ are gradient and divergence in Riemannian manifold, and $\Delta_M$ is the Laplace-Beltrami operator generalizing the Euclidean Laplacian $\Delta$ to Riemannian manifold. More details can be found in Appendix. The stationary solution of equation (3) is $e^{-f(x)}$ that minimizes the entropy regularized functional $\mathcal{L}(\rho)$, and then the optimization problem over the space of densities boils down to track the evolution of $\rho(x,t)$ that is defined by equation (3).

**Coordinate-independent Langevin equation.** In order to implement the aforementioned evolution of $\rho(x,t)$ in Euclidean space, one can simulate the stochastic process $\{X_t\}_{t\geq 0}$ defined by the Langevin equation: $dX_t = -\nabla f(X_t)dt + \sqrt{2}dB_t$, where $B_t$ is the standard Brownian motion and $X_t$ has $\rho(x,t)$ as its density function. In contrast to the Euclidean case, we need a coordinate-independent formulation of Langevin equation, e.g. Batrouni et al. [1986]. This is derived by expanding the Fokker-Planck equation (3) in a local coordinate system and is written in the following form:

$$dX_t = F(X_t)dt + \sqrt{2g^{-1}}dB_t \tag{4}$$

where $F(X_t)$ is a vector with $i$'th component $F_i = -\sum_j g^{ij}\frac{\partial f}{\partial x_j} + \frac{1}{\sqrt{|g|}}\sum_j \frac{\partial}{\partial x_j}\left(\sqrt{|g|}g^{ij}\right)$ and $|g|$ is the determinant of metric matrix $g_{ij}$. Note that this local form indicate the fact from Fokker-Planck equation that the process $\{X_t\}_{t\geq 0}$ is the negative gradient of $f$ followed by a manifold Brownian motion. The rate of convergence we are interested in is the classic Euler-Maruyama discretization scheme in manifold setting, i.e. compute the vector in tangent space and project it onto the base manifold through exponential map. So the discretization error consists of two parts: one is from considering $\mathrm{grad}f(x_t)$ as constant in a neighborhood of $x_t$, and the other is from the approximation of a curved neighborhood of $x_t$ with the tangent space at $x_t$. The main task in the proofs of Theorem 4.3 and **??** is to bound the aforementioned two parts of errors and compare with the density evolving along continuous time Langevin equation.

### 4.2 Convergence Analysis

We state some assumptions before presenting main theorems.

**Assumption 1.** *$M$ is a closed manifold.*

It means $M$ is compact and has no boundary, Definition 3.2. This assumption is essentially used to make the boundary integral on $\partial M$, i.e. $\int_{\partial M} \log \frac{\rho_t}{\nu} \langle \rho_t \mathrm{grad} f + \mathrm{grad} \rho_t, \mathbf{n} \rangle dx$ in the proof of Lemma 4.1, to vanish, see Appendix. This assumption can be relaxed to open manifold by assuming the integral decreases fast as $x$ approaches the infinity.

**Assumption 2.** $f(x)$ *is differentiable on* $M$.

An immediate consequence by combining Assumption 1 and 2 is that there exists a number $L > 0$, such that the Riemannian gradient $\mathrm{grad} f$ of $f$ is $L$-Lipschitz (Definition 3.8) due to the compactness of $M$. Another crucial property used in the proof is that the target distribution $e^{-f}$ satisfies the log-Sobolev inequality, and this can also be derived by compactness of $M$.

**Assumption 3.** *We next assume the existence of constants shown in the convergence result. Let the joint distribution* $p_{0t}(x_0, x)$ *be differentiable and assume that* $\frac{\frac{\partial^2 p_{0t}(x_0, x)}{\partial x_i \partial x_j} \log \frac{p_t}{\nu}}{p_{0t}(x_0, x)}$ *is bounded by* $K_2$ *and* $\frac{\left| \frac{\partial p_{0t}(x_0, x)}{\partial x_i} \right| \log \frac{p_t}{\nu}}{p_{0t}(x_0, x)}$ *is bounded by* $K_3$.

Since the prerequisite of convergence of GLA is the convergence of the continuous time Langevin equation, we show the KL divergence between $\rho_t$ and $\nu$ converges along the continuous time Riemannian Langevin equation. The proof is completed by the following lemma showing that $H(\rho_t | \nu)$ decreases since $\frac{d}{dt} H(\rho_t | \nu) < 0$ for all $\rho_t \neq \nu$. Based on the analysis of the previous section, it suffices to track the evolution of $\rho_t$ according to the Fokker-Planck equation (3).

**Lemma 4.1.** *Suppose* $\rho_t$ *evolves following the Fokker-Planck equation (3), then*

$$\frac{d}{dt} H(\rho_t | \nu) = - \int_M \rho_t(x) \left\| \mathrm{grad} \log \frac{\rho_t(x)}{\nu(x)} \right\|^2 dx$$

*where* $dx$ *is the Riemannian volume element.*

The proof is a straightforward calculation of the time derivative of $H(\rho_t | \nu)$, followed by the expression of $\frac{\partial \rho_t}{\partial t}$ in equation (3), i.e.

$$\frac{d}{dt} H(\rho_t | \nu) = \int_M \frac{\partial \rho_t}{\partial t} \log \frac{\rho_t}{\nu} dx = \int_M \mathrm{div}(\rho_t \mathrm{grad} f + \mathrm{grad} \rho_t) \log \frac{\rho_t}{\nu} dx \tag{5}$$

$$= - \int_M \rho_t \left\| \mathrm{grad} \log \frac{\rho_t}{\nu} \right\|^2 dx + \int_{\partial M} \log \frac{\rho_t}{\nu} \langle \rho_t \mathrm{grad} f + \mathrm{grad} \rho_t, \mathbf{n} \rangle dx \tag{6}$$

The result follows from integration by parts and the Assumption 1. Details are left in Appendix.

Since $M$ is compact, there exists a constant $\alpha > 0$ such that the log-Sobolev inequality (LSI) holds. So we can get the following convergence of KL divergence for continuous Langevin dynamics immediately.

**Theorem 4.2.** *Suppose* $\nu$ *satisfies LSI with constant* $\alpha > 0$. *Then along the Riemannian Langevin equation, i.e. the SDE (4) in local coordinate systems, the density* $\rho_t$ *satisfies*

$$H(\rho_t | \nu) \leq e^{-2\alpha t} H(\rho_0 | \nu).$$

The following theorem shows that the KL divergence $H(\rho_k | \nu)$ decreases geometrically along the GLA dynamics.

**Theorem 4.3.** *Suppose* $M$ *is a compact manifold without boundary and* $R$ *is the Riemann curvature,* $\nu = e^{-f}$ *a density on* $M$ *with* $\alpha > 0$ *the log-Sobolev constant. Then there exists a global constant* $K_2, K_3, K_4, C$, *such that for any* $x_0 \sim \rho_0$ *with* $H(\rho_0 | \nu) \leq \infty$, *the iterates* $x_k \sim \rho_k$ *of GLA with stepsize* $\epsilon \leq \min\{\frac{\alpha}{4L\sqrt{L^2 + n^2 K_2 C}}, \frac{2nL^2 + 2n^3 K_2 C + nK_3 K_4}{2(nL^3 + n^3 K_2 CL)}, \frac{1}{2L}, \frac{1}{2\alpha}\}$ *satisfty*

$$H(p_k | \nu) \leq e^{-\alpha k \epsilon} H(p_0 | \nu) + \frac{16\epsilon}{3\alpha}(2nL^2 + 2n^3 K_2 C + nK_3 K_4)$$

The convergence of KL divergence implies the convergence of Wasserstein distance.

**Proposition 4.4.** *For the closed manifold* $M$ *with a density* $\nu = e^{-f(x)}$, *the iterates* $x_k \sim \rho_k$ *of GLA with a properly chosen stepsize satisfy*

$$W_2(\rho_k, \nu)^2 \leq \frac{2}{\alpha} e^{-\alpha k \epsilon} H(p_0 | \nu) + \frac{32\epsilon}{3\alpha^2}(2nL^2 + 2n^3 K_2 C + nK_3 K_4).$$

*Proof.* It is an immediate consequence from the convergence of KL divergence that the Wasserstein distance $W_2(\rho_k, \nu)$ converges rabidly since log-Sobolev inequality implies Talagrand inequality (Talagrand [1996], Otto and Villani [2000]). Since $\nu$ satisfies log-Sobolev inequality, then we have

$$W_2(\rho_k|\nu)^2 \leq \frac{2}{\alpha} H(\rho_k|\nu) \tag{7}$$

$$\leq \frac{2}{\alpha} \left( e^{-\alpha k \epsilon} H(p_0|\nu) + \frac{16\epsilon}{3\alpha} (2nL^2 + 2n^3 K_2 C + n K_3 K_4) \right) \tag{8}$$

$$= \frac{2}{\alpha} e^{-\alpha k \epsilon} H(p_0|\nu) + \frac{32\epsilon}{3\alpha^2} (2nL^2 + 2n^3 K_2 C + n K_3 K_4) \tag{9}$$

and the proof completes. $\square$

## 5   Conclusion

In this paper we focus on the problem of sampling from a distribution on a Riemannian manifold and propose the Geodesic Langevin Algorithm. GLA modifies the Riemannian Langevin algorithm by using exponential map so that the algorithm is defined globally. By leveraging the geometric meaning of GLA, we provide a non-asymptotic convergence guarantee in the sense that the KL divergence (as well as the Wasserstein distance) decreases fast along the iterations of GLA. By assuming that we have full access to the geometric data (curvature, geodesic distance, Ricci curvature, diameter, etc) of the manifold, we can control the bias between the stationary distribution of GLA and the target distribution to be arbitrarily small through the choice of stepsize.

## 6   Acknowledgements

Xiao Wang would like to acknowledge the NRF-NRFFAI1-2019-0003, SRG ISTD 2018 136 and NRF2019-NRF-ANR095 ALIAS grant. Qi Lei is supported by the NSF under Grant #2030859 to the Computing Research Association for the CIFellows Project.

## 7   Broader Impact

Our results is motivated by variant applications of sampling algorithms in machine learning. Generating random sample especially in higher dimension with good convergence and error bound can be used in fast integration and volume computation. With the development of continuous game theory and GANs in recent years, our results have potential impact to solve the Nash equilibrium of the games with continuous strategy space.

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
