[Supplementary Material]

# Supplementary Material for
# Fast Convergence of Langevin Dynamics on Manifold: Geodesics meet Log-Sobolev

**Xiao Wang**
SUTD
xiao_wang@sutd.edu.sg

**Qi Lei**
Princeton
qilei@princeton.edu

**Ioannis Panageas**
UC Irvine
ipanagea@uci.edu

## A   More background

### A.1   Calculus on manifold

**Definition A.1** (Levi-Civita Connection)**.** Let $(M, g)$ be a Riemannian manifold. An affine connection is said to be the Levi-Civita connection if it is torsion-free. i.e.

$$\nabla_X Y - \nabla_Y X = [X, Y]$$

for every pair of vector fields $X, Y$ on $M$ and preserves the metric i.e.

$$\nabla g = 0.$$

**Definition A.2** (Riemannian Volume)**.** Let $(M, g)$ be an orientable Riemannian manifold. The volume form on the manifold in local coordinates is given as

$$d\text{Vol} = \sqrt{\det(g)} dx_1 \wedge ... \wedge dx_n.$$

We denote $|g| = \det(g)$ and $dx = d\text{Vol}$ (if no ambiguities caused) for short throughout following context.

The following Theorem is used to guarantee the exponential map is defined on the whole tangent space, which is equivalent to require $M$ to be complete. This property is satisfied in our setting for $M$ to be compact without boundary.

**Theorem A.3** (Hopf-Rinow)**.** *Let $(M, g)$ be a connected Riemannian manifold. Then the followings are equivalent.*

1. *The closed and bounded subsets of $M$ are compact.*

2. *$M$ is a complete metric space.*

3. *$M$ is geodesically complete: for every point $x \in M$, the exponential map $\text{Exp}_x$ is defined on the entire tangent space $T_x M$.*

The notion of differential operators, e.g. gradient, divergence and Laplacian for the differentiable functions and vector fields on Euclidean space can be generalized to Riemannian manifold. In local coordinate system, $\{\partial_i = \frac{\partial}{\partial x_i} : i \in [n]\}$ is a basis of the tangent space $T_x M$. Denote $g_{ij}$ the metric matrix, $g^{ij}$ the inverse of $g_{ij}$ and $|g| = \det g_{ij}$ the determinant of matrix $g_{ij}$. Let $f$ and $V$ be differentiable function and vector field on $M$, then the Riemannian gradient of $f$ and the divergence of $V$ are written as

$$\text{grad} f = \sum_{i,j} g^{ij} \frac{\partial f}{\partial x_i} \partial_i \quad \text{and} \quad \text{div} V = \frac{1}{\sqrt{|g|}} \sum_i \frac{\partial}{\partial x_i} \left( \sqrt{|g|} V_i \right)$$

where $V_i$ is the $i$-th component of $V$.

The Laplace-Beltrami operator $\Delta_M$ acting on $f$ is defined to be the divergence of the gradient of $f$, i.e.

$$\Delta_M f = \text{div}(\text{grad} f) = \frac{1}{\sqrt{|g|}} \sum_i \frac{\partial}{\partial x_i} \left( \sqrt{|g|} \sum_j g^{ij} \frac{\partial f}{\partial x_j} \right).$$

In Euclidean space, $\Delta_M$ boils down to the classic Laplacian $\Delta f = \nabla \cdot (\nabla f)$.

The following integration by parts formulas are used in proof of main lemmas. Let $M$ be a compact oriented Riemannian manifold of dimension $n$ with boundary $\partial M$. Let $X$ be a vector field on $M$. The integration by parts is given by

$$\int_M \langle \text{grad} f, X \rangle = - \int_M f \text{div} X + \int_{\partial M} f \langle X, n \rangle$$

or Green's formula

$$\int_M (f \Delta_M g - g \Delta_M f) = - \int_{\partial M} \left( f \frac{\partial g}{\partial n} - g \frac{\partial f}{\partial n} \right)$$

If $\partial M$ is empty or the vector field $X$ decay sufficiently fast at infinity of $M$ provided $M$ is open, we have

$$\int_M \langle \text{grad} f, X \rangle = - \int_M f \text{div} X.$$

**Definition A.4** (First eigenvalue of Laplacian). The first eigenvalue $\lambda_1 \geq 0$ of the Laplacian operator on $M$ is defined to be

$$\lambda_1 = \inf_{f \in C_c^\infty} \left\{ \frac{\int_M \|\text{grad} f\|^2 dx}{\int_M \|f\|^2 dx} \right\}.$$

## A.2  Stochastic analysis on manifold

Recall that the standard Brownian motion in $\mathbb{R}^n$ is a random process $\{X_t\}_{t \geq 0}$ whose density evolves according to the diffusion equation

$$\frac{\partial \rho(x,t)}{\partial t} = \frac{1}{2} \Delta \rho(x,t).$$

Similarly, the Brownian motion in manifold $M$ is $M$-valued random process $\{W_t\}_{t \geq 0}$ whose density function evolves according to the diffusion equation with respect to Laplace-Beltrami operator which is the counterpart of the Laplace operator on Euclidean space.

$$\frac{\partial \rho(x,t)}{\partial t} = \frac{1}{2} \Delta_M \rho(x,t).$$

In local coordinate, the Laplace-Beltrami is written as

$$\Delta_M = \sum_{i,j} g^{ij} \frac{\partial^2}{\partial x_i \partial x_j} + \sum_i b_i \frac{\partial}{\partial x_i},$$

where

$$b_i = \sum_j \frac{1}{\sqrt{|g|}} \frac{\partial}{\partial x_j} \left( \sqrt{|g|} g^{ij} \right) = \sum_{j,k} g^{jk} \Gamma^i_{jk}. \tag{1}$$

We can construct Brownian motion in the local coordinate as the solution of the stochastic differential equation for a process $\{X_t\}_{t \geq 0}$:

$$dX_t = \frac{1}{2} b(X_t) dt + \sigma(X_t) dB_t$$

where the component $b_i(X_t)$ of $b(X_t)$ is given by (1) and $\sigma = (\sigma_{ij})$ is the unique symmetric square root of $g^{-1} = (g^{ij})$.

# B  Derivation of the GLA

In this section, we give detailed explanation on that the Riemannian Langevin algorithm, as a stochastic process, captures the dynamics of the evolution of the density function for the stochastic process. The derivation is firstly to write the diffusion equation in local coordinate system of the manifold, and then compare the corresponding terms to the Fokker-Planck equation related to stochastic differential equation that gives insight to the local expression of Riemannian Langevin algorithm. In order to do this, recall that the density $e^{-f}$ on $M$ is the stationary solution of the PDE

$$\frac{\partial \rho_t}{\partial t} = \mathrm{div}\left(\rho_t \mathrm{grad} f + \mathrm{grad} \rho_t\right). \tag{2}$$

Using the local expression of Riemannian gradient and divergence operator, this PDE can be written as

$$\frac{\partial \rho_t}{\partial t} = \frac{1}{\sqrt{|g|}} \sum_{i=1}^{n} \frac{\partial}{\partial x_i}\left(\sqrt{|g|}\left(\sum_j g^{ij}\frac{\partial f}{\partial x_j}\right)\rho_t + \sqrt{|g|}\sum_j g^{ij}\frac{\partial \rho_t}{\partial x_j}\right) \tag{3}$$

$$= \frac{1}{\sqrt{|g|}} \sum_i \frac{\partial}{\partial x_i}\left(\left(\sum_j g^{ij}\frac{\partial f}{\partial x_j} - \frac{1}{\sqrt{|g|}}\sum_j \frac{\partial}{\partial x_j}\left(\sqrt{|g|}g^{ij}\right)\right)\sqrt{|g|}\rho_t\right) \tag{4}$$

$$+ \frac{1}{\sqrt{|g|}} \sum_{i,j} \frac{\partial^2}{\partial x_i \partial x_j}\left(g^{ij}\sqrt{|g|}\rho_t\right) \tag{5}$$

Denoting $\tilde{\rho}_t = \sqrt{|g|}\rho_t$, we have the Fokker-Planck equation of density in Euclidean space as follows,

$$\frac{\partial \tilde{\rho}_t}{\partial t} = -\sum_i \frac{\partial}{\partial x_i}\left(\left(\frac{1}{\sqrt{|g|}}\sum_j \frac{\partial}{\partial x_j}\left(\sqrt{|g|}g^{ij}\right) - \sum_j g^{ij}\frac{\partial f}{\partial x_j}\right)\tilde{\rho}_t\right) + \sum_{i,j} \frac{\partial^2}{\partial x_i \partial x_j}\left(g^{ij}\tilde{\rho}_t\right). \tag{6}$$

Since for any stochastic differential equation of the form

$$dX_t = F(X_t, t)dt + \sigma(X_t, t)dB_t$$

the density $p_t$ for $X_t$ satisfies

$$\frac{\partial p(x,t)}{\partial t} = -\sum_{i=1}^{n} \frac{\partial}{\partial x_i}\left(F_i(x,t)p(x,t)\right) + \sum_{i=1}^{n}\sum_{j=1}^{n} \frac{\partial^2}{\partial x_i \partial x_j}\left(A_{ij}(x,t)p(x,t)\right) \tag{7}$$

where $A = \frac{1}{2}\sigma\sigma^\top$, i.e. $A_{ij} = \frac{1}{2}\sum_{k=1}^{n}\sigma_{ik}(x,t)\sigma_{jk}(x,t)$. Compare equations (6) and (7), we have the drift and diffusion terms in local coordinate systems are given by

$$F_i(x_t) = -\sum_j g^{ij}\frac{\partial f}{\partial x_j} + \frac{1}{\sqrt{|g|}}\sum_j \frac{\partial}{\partial x_j}\left(\sqrt{|g|}g^{ij}\right)$$

and

$$\sigma(x_t) = \sqrt{2(A_{ij})} = \sqrt{2(g^{ij})} = \sqrt{2g^{-1}}.$$

So the local Langevin equation is

$$dX_t = F(X_t)dt + \sqrt{2g^{-1}}dB_t. \tag{8}$$

This equation describes infinitesimal evolution of $X_t$, which can be seen as a process in the tangent space of $M$. The Riemannian Langevin algorithm is the classic Euler-Maruyama discretization in the tangent space, i.e., by letting $X_t$ move in the tangent space for a positive time interval $t \in [0, \epsilon]$ with the drift and diffusion at current location. Suppose the initial point is $x_0$, the tangent vector is

$$\epsilon F(x_0) + \sqrt{2\epsilon g^{-1}(x_0)}\xi_0$$

where $\xi_0 \sim \mathcal{N}(0, I)$ is the standard Gaussian noise. Then the updated point is obtained by mapping the vector to the base manifold via exponential map,

$$x_1 = \mathrm{Exp}_{x_0}\left(\epsilon F(x_0) + \sqrt{2\epsilon g^{-1}(x_0)}\xi_0\right).$$

Renaming $x_k = x_0$ and $x_{k+1} = x_1$, we have the general form

$$x_{k+1} = \operatorname{Exp}_{x_k}\left(\epsilon F(x_k) + \sqrt{2\epsilon g^{-1}(x_k)}\xi_0\right).$$

We give the expression of the algorithm in normal coordinate, for convenience in part of the proofs of main theorems.

For any manifold $M$, and $x \in M$, $T_x M$ is isomorphic to $\mathbb{R}^n$, $\exp_x^{-1}$ gives a local coordinate system of $M$ around $x$. This is called the normal coordinates at $x$. The following lemmas are from Lee-Vampalar

**Lemma B.1.** *In normal coordinate, we have*

$$g_{ij}(x) = \delta_{ij} - \frac{1}{3}\sum_{kl} R_{ikjl}(x)x^k x^l + O(|x|^3).$$

Under normal coordinate, the RLA can be written as

$$x_{t+1} = \operatorname{Exp}_{x_t}(-\epsilon\nabla f(x_t) + \sqrt{2\epsilon}\xi_0).$$

Note that the expression in the tangent space is exactly the same as unadjusted Langevin algorithm in Euclidean space.

# C   Missing proofs of Section 4

## C.1   Proof of Theorem 4.2

In this section, we proof that the KL divergence decreases along the process evolving following Riemannian Langevin equation.

Firstly, need show that according to the SDE on manifold in local chart, the density function evolves according to Fokker-Planck/diffusion equation on this manifold.

**Lemma 4.1.** Suppose $\rho_t$ evolves following the Fokker-Planck equation (3), then

$$\frac{d}{dt}H(\rho_t|\nu) = -\int_M \rho_t(x)\left\|\operatorname{grad}\log\frac{\rho_t(x)}{\nu(x)}\right\|^2 dx$$

where $dx$ is the Riemannian volume element.

*Proof.* Since

$$\int_M \rho_t\frac{\partial}{\partial t}\log\frac{\rho_t}{\nu}dx = \int_M \rho_t\frac{\partial}{\partial t}(\log\rho_t + f)dx \tag{9}$$

$$= \int_M \frac{\partial\rho_t}{\partial t}dx \tag{10}$$

$$= \frac{d}{dt}\int_M \rho_t dx = 0, \tag{11}$$

we have

$$\frac{d}{dt}H(\rho_t|\nu) = \frac{d}{dt}\int_M \rho_t\log\frac{\rho_t}{\nu}dx \tag{12}$$

$$= \int_M \frac{d}{dt}\left(\rho_t\log\frac{\rho_t}{\nu}\right)dx \tag{13}$$

$$= \int_M \frac{\partial\rho_t}{\partial t}\log\frac{\rho_t}{\nu}dx + \int_M \rho_t\frac{\partial}{\partial t}\log\frac{\rho_t}{\nu}dx \tag{14}$$

$$= \int_M \frac{\partial\rho_t}{\partial t}\log\frac{\rho_t}{\nu}dx. \tag{15}$$

Plug in with diffusion equation

$$\frac{\partial\rho_t}{\partial t} = \operatorname{div}(\rho_t\operatorname{grad}f + \operatorname{grad}\rho_t)$$

and apply integration by parts, we obtain

$$\frac{d}{dt}H(\rho_t|\nu) = \int_M \text{div}(\rho_t \text{grad} f + \text{grad}\rho_t) \log \frac{\rho_t}{\nu} dx \tag{16}$$

$$= -\int_M \rho_t \left\|\text{grad} \log \frac{\rho_t}{\nu}\right\|^2 dx \tag{17}$$

$$+ \int_{\partial M} \log \frac{\rho_t}{\nu} \langle \rho_t \text{grad} f + \text{grad}\rho_t, n\rangle dx \tag{18}$$

Since $M$ is compact and has no boundary, the boundary integral equals to zero, then we have

$$\frac{d}{dt}H(\rho_t|\nu) = -\int_M \rho_t \left\|\text{grad} \log \frac{\rho_t}{\nu}\right\|^2 dx$$

$\square$

**Theorem 4.2.** Suppose $\nu$ satisfies LSI with constant $\alpha > 0$. Then along the Riemannian Langevin equation, i.e. the SDE (4) in local coordinate systems, the density $\rho_t$ satisfies

$$H(\rho_t|\nu) \leq e^{-2\alpha t}H(\rho_0|\nu).$$

*Proof.* By LSI, we have

$$\frac{d}{dt}H(\rho_t|\nu) \leq -2\alpha H(\rho_t|\nu),$$

multiplying both sides by $e^{\alpha t}$,

$$e^{2\alpha t}\frac{d}{dt}H(\rho_t|\nu) \leq -2\alpha e^{2\alpha t}H(\rho_t|\nu)$$

and then

$$e^{2\alpha t}\frac{d}{dt}H(\rho_t|\nu) + 2\alpha e^{2\alpha t}H(\rho_t|\nu) = \frac{d}{dt}e^{2\alpha t}H(\rho_t|\nu) \leq 0.$$

Integrating for $0 \leq t \leq s$, the result holds as

$$e^{2\alpha s}H(\rho_s|\nu) - H(\rho_0|\nu) = \int_0^s \frac{d}{dt}e^{2\alpha t}H(\rho_t|\nu) \leq 0.$$

Rearranging and renaming $s$ by $t$, we conclude

$$H(\rho_t|\nu) \leq e^{-2\alpha t}H(\rho_t|\nu)$$

$\square$

### C.2  Proof of Theorem 4.3

**Lemma C.1.** *Assume $\nu = e^{-f}$ is L-smooth. Then*

$$\mathbb{E}_\nu[\|\text{grad} f\|^2] \leq nL.$$

*Proof.* Since

$$\mathbb{E}_\nu[\|\text{grad} f\|^2] = \int_M \langle \text{grad} f, \text{grad} f\rangle e^{-f} dx = -\int_M \langle \text{grad} e^{-f}, \text{grad} f\rangle dx,$$

where $dx$ is the Riemannian volume element. Integration by parts on manifold gives the following

$$-\int_M \langle \text{grad} e^{-f}, \text{grad} f\rangle dx = \int_M e^{-f}\Delta_M f dx - \int_{\partial M} e^{-f}\langle \text{grad} f, n\rangle ds$$

where $ds$ is the area element on $\partial M$. By the assumption that $M$ is boundaryless, the integral on the boundary is 0. By the assumption $\text{Hess} f$ is $L$-smooth and the fact that $\text{Hess} f \geq \frac{1}{n}\Delta_M f$, we conclude $\mathbb{E}_\nu[\|\text{grad} f\|^2] \leq nL$ $\square$

**Lemma C.2.** *Suppose $\nu$ satisfies Talagrand inequality with constant $\alpha > 0$ and L-smooth. Then for any $\rho$,*

$$\mathbb{E}_\rho[\|\mathrm{grad}f\|^2] \leq \frac{4L^2}{\alpha}H(\rho|\nu) + 2nL.$$

*Proof.* Let $x \sim \rho$ and $y \sim \nu$ with optimal coupling $(x, y)$ so that

$$\mathbb{E}[d(x,y)^2] = W_2(\rho,\nu)^2.$$

$\mathrm{grad}f$ is $L$-Lipschitz from the assumption that $f$ is $L$-smooth. So we have the following inequality:

$$\|\mathrm{grad}f(x)\| \leq \|\mathrm{grad}f(x) - \Gamma_y^x \mathrm{grad}f(y)\| + \|\Gamma_y^x \mathrm{grad}f(y)\| \tag{19}$$
$$\leq Ld(x,y) + \|\Gamma_y^x \mathrm{grad}f(y)\| \tag{20}$$
$$= Ld(x,y) + \|\mathrm{grad}f(y)\| \tag{21}$$

where the equality follows from that parallel transport is an isometry. The same arguments as V-W gives

$$\|\mathrm{grad}f(x)\|^2 \leq (Ld(x,y) + \|\mathrm{grad}f(y)\|)^2 \leq 2Ld(x,y)^2 + 2\|\mathrm{grad}f(y)\|^2$$

and

$$\mathbb{E}_\rho[\|\mathrm{grad}f(x)\|^2] \leq 2L^2\mathbb{E}[d(x,y)^2] + 2\mathbb{E}_\nu[\|\mathrm{grad}f(y)\|^2] \tag{22}$$
$$= 2L^2 W_2(\rho,\nu)^2 + 2\mathbb{E}_\nu[\|\mathrm{grad}f(y)\|^2]. \tag{23}$$

By Talagrand inequality and previous lemma, the result follows. $\square$

**Lemma C.3.** *Suppose $\nu$ satisfies LSI with constant $\alpha > 0$ and is L-smooth. If $\epsilon$ small enough, then along each step,*

$$H(p_\epsilon|\nu) \leq e^{-\alpha\epsilon}H(p_0|\nu) + 4\epsilon^2(2nL^2 + 2n^3 K_2 C + nK_3 K_4)$$

*for small $\epsilon$, and*

$$H(p_{k+1}|\nu) \leq e^{-\alpha\epsilon}H(p_k|\nu) + 4\epsilon^2(2nL^2 + 2n^3 K_2 C + nK_3 K_4).$$

*for all $k \in \mathbb{N}$.*

*Proof.* According to [**?** ], the exponential map $\mathrm{Exp}_x$ is a diffeomorphism on almost all the manifold, i.e. let $\bar{U}_x$ be the closed set of vectors in $T_x M$ for which $\gamma(t) = \mathrm{Exp}_x(tv), t \in [0,1]$ is length minimizing, and $U_x$ be the interior and $\partial \bar{U}_x$ be its boundary. Then the exponential map is a diffeomorphism on $U_x$ and $\mathrm{Exp}_x(\partial \bar{U}_x)$ has measure zero.

In normal coordinates, the discretized SDE has the form of

$$dx_t = -\mathrm{grad}f(x_0)dt + \sqrt{2g^{-1}(x_0)}dB_t$$

and the Fokker-Planck equation of this SDE is

$$\frac{\partial p_{t|0}(x_t|x_0)}{\partial t} = \sum_i \frac{\partial}{\partial x_i}((\sum_j g^{ij}\frac{\partial f}{\partial x_j}(x_0))p_{t|0}(x_t|x_0)) + \sum_{i,j}\frac{\partial^2}{\partial x_i \partial x_j}g^{-1}(x_0)p_{t|0}(x_t|x_0) \tag{24}$$

$$= \mathrm{div}(p_{t|0}(x_t|x_0)\mathrm{grad}f(x_0)) + \sum_{i,j}g^{-1}\frac{\partial^2}{\partial x_i \partial x_j}p_{t|0}(x_t|x_0) + \sum_i b_i\frac{\partial}{\partial x_i}p_{t|0}(x_t|x_0) \tag{25}$$

$$+ \sum_{i,j}\frac{\partial^2}{\partial x_i \partial x_j}g^{-1}(x_0)p_{t|0}(x_t|x_0) - (\sum_{i,j}g^{-1}\frac{\partial^2}{\partial x_i \partial x_j}p_{t|0}(x_t|x_0) + \sum_i b_i\frac{\partial}{\partial x_i}p_{t|0}(x_t|x_0)) \tag{26}$$

$$= \mathrm{div}(p_{t|0}(x_t|x_0)\mathrm{grad}f(x_0)) + \Delta_M p_{t|0}(x_t|x_0) \tag{27}$$

$$+ \sum_{i,j}(g^{ij}(x_0) - g^{ij}(x_t))\frac{\partial^2}{\partial x_i \partial x_j}p_{t|0}(x_t|x_0) - \sum_i b_i\frac{\partial}{\partial x_i}p_{t|0}(x_t|x_0) \tag{28}$$

$$\frac{\partial p_t(x)}{\partial t} = \int_{\mathbb{R}^n} \frac{\partial p_{t|0}(x|x_0)}{\partial t} p_0(x_0) \sqrt{|g|} dx_0 \tag{29}$$

$$= \int_{\mathbb{R}^n} \left( \sum_i \frac{\partial}{\partial x_i}((\sum_j g^{ij} \frac{\partial f}{\partial x_j}(x_0)) p_{t|0}(x_t|x_0)) + \sum_{i,j} \frac{\partial^2}{\partial x_i \partial x_j} g^{-1}(x_0) p_{t|0}(x_t|x_0) \right) p_0(x_0) \sqrt{|g|} dx_0 \tag{30}$$

$$= \int_{\mathbb{R}^n} \left( \mathrm{div}(p_{t|0}(x|x_0) \mathrm{grad} f(x_0)) + \Delta_M p_{t|0}(x|x_0) \right) p_0(x_0) \sqrt{|g|} dx_0 \tag{31}$$

$$+ \int_{\mathbb{R}^n} \left( \sum_{i,j} (g^{ij}(x_0) - g^{ij}(x)) \frac{\partial^2}{\partial x_i \partial x_j} p_{t|0}(x|x_0) \right) p_0(x_0) \sqrt{|g|} dx_0 \tag{32}$$

$$- \int_{\mathbb{R}^n} \left( \sum_i b_i \frac{\partial}{\partial x_i} p_{t|0}(x|x_0) \right) p_0(x_0) \sqrt{|g|} dx_0 \tag{33}$$

$$\int_{\mathbb{R}^n} \left( \mathrm{div}(p_{t|0}(x|x_0) \mathrm{grad} f(x_0)) + \Delta_M p_{t|0}(x|x_0) \right) p_0(x_0) \sqrt{|g|} dx_0 \tag{34}$$

$$= \int_{\mathbb{R}^n} \mathrm{div}(p_{t0}(x, x_0) \mathrm{grad} f(x_0)) \sqrt{|g|} dx_0 + \Delta_M p_t(x) \tag{35}$$

$$= \mathrm{div} \left( p_t(x) \int_{\mathbb{R}^n} p_{0|t}(x_0|x) \mathrm{grad} f(x_0) dx_0 \right) + \Delta_M p_t(x) \tag{36}$$

$$= \mathrm{div} \left( p_t(x) \mathbb{E}_{p_{0|t}}[\mathrm{grad} f(x_0)|x_t = x] \right) + \Delta_M p_t(x) \tag{37}$$

$$\int_{\mathbb{R}^n} \left( \sum_{i,j} (g^{ij}(x_0) - g^{ij}(x)) \frac{\partial^2}{\partial x_i \partial x_j} p_{t|0}(x|x_0) \right) p_0(x_0) \sqrt{|g|} dx_0 \tag{38}$$

$$= \sum_{i,j} \int_{\mathbb{R}^n} (g^{ij}(x_0) - g^{ij}(x)) \frac{\partial^2}{\partial x_i \partial x_j} p_{t|0}(x|x_0) p_0(x_0) \sqrt{|g|} dx_0 \tag{39}$$

$$\leq \sum_{i,j} \int_{\mathbb{R}^n} |g^{ij}(x_0) - g^{ij}(x)| \cdot \left| \frac{\partial^2}{\partial x_i \partial x_j} p_{t|0}(x|x_0) p_0(x_0) \right| \sqrt{|g|} dx_0 \tag{40}$$

$$\leq \sum_{i,j} \int_{\mathbb{R}^n} O(\|x - x_0\|^2) \left| \frac{\partial^2}{\partial x_i \partial x_j} p_{t|0}(x|x_0) p_0(x_0) \right| \sqrt{|g|} dx_0 \tag{41}$$

$$= \sum_{i,j} \int_{\mathbb{R}^n} O(\|x - x_0\|^2) \left| \frac{\partial^2}{\partial x_i \partial x_j} p_{0t}(x_0, x) \right| \sqrt{|g|} dx_0 \tag{42}$$

$$\leq n^2 K_1 \int_{\mathbb{R}^n} O(\|x - x_0\|^2) \sqrt{|g|} dx_0 \tag{43}$$

where $K_1$ is the upper bound of $\left| \frac{\partial^2}{\partial x_i \partial x_j} p_{0t}(x_0, x) \right|$

$$\int_{\mathbb{R}^n} \left( \int_{\mathbb{R}^n} \left( \sum_{i,j} (g^{ij}(x_0) - g^{ij}(x)) \frac{\partial^2}{\partial x_i \partial x_j} p_{t|0}(x|x_0) \right) p_0(x_0) \sqrt{|g|} dx_0 \right) \log \frac{p_t}{\nu} \sqrt{|g|} dx \tag{44}$$

$$\leq \sum_{ij} \int_{\mathbb{R}^n} \left( \int_{\mathbb{R}^n} |g^{ij}(x_0) - g^{ij}(x)| \left| \frac{\partial^2}{\partial x_i \partial x_j} p_{0t}(x_0, x) \right| \sqrt{|g|} dx_0 \right) \left| \log \frac{p_t}{\nu} \right| \sqrt{|g|} dx \tag{45}$$

Let

$$\tilde{p}(x_0, x) = \frac{\frac{\partial^2 p_{t0}(x_0,x)}{\partial x_i \partial x_j} \log \frac{p_t}{\nu}}{p_{t0}(x_0,x)}$$

and assume that $|\tilde{p}(x_0,x)|$ is bounded by $K_2$. Then (44) is bounded by $n^2 K_2 \mathbb{E}_{p_{t0}}\left[O\left(\left\|-t\mathrm{grad}f(x_0) + \sqrt{2t}z\right\|^2\right)\right]$.

$$\int_{\mathbb{R}^n}\left(\int_{\mathbb{R}^n}\left(\sum_{ij} b_i \frac{\partial p_{0t}(x_0,x)}{\partial x_i}\right)\sqrt{|g(x_0)|}dx_0\right)\log\frac{p_t}{\nu}\sqrt{|g(x)|}dx \tag{46}$$

$$= \sum_i \int_{\mathbb{R}^n \times \mathbb{R}^n} b_i(x)\frac{\partial p_{0t}(x_0,x)}{\partial x_i}\log\frac{p_t}{\nu}d(x_0 \times x) \tag{47}$$

$$\leq K_3 \int_{\mathbb{R}^n \times \mathbb{R}^n} |b_i(x)|\, p_{0t}(x_0,x)d(x_0 \times x) \ \ \left(\text{suppose } K_3 \geq \frac{\left|\frac{\partial p_{0t}(x_0,x)}{\partial x_i}\right|\log\frac{p_t}{\nu}}{p_{0t}(x_0,x)}\right) \tag{48}$$

$$= K_3 \sum_i \mathbb{E}_{p_{0t}}[|b_i(x)|] \tag{49}$$

$$= K_3 \sum_i \mathbb{E}_{p_{0t}}\left[\left|b_i(x_0 - t\mathrm{grad}f(x_0) + \sqrt{2t}z)\right|\right] \tag{50}$$

$$b_i(x_0 - t\mathrm{grad}f(x_0) + \sqrt{2t}z) \tag{51}$$

$$= b_i(x_0) - t\langle\nabla b_i(x_0), \nabla f(x_0)\rangle + \sqrt{2t}\langle\nabla b_i(x_0), z\rangle + t\langle z, \nabla^2 b_i(x_0)z\rangle. \tag{52}$$

and then

$$\mathbb{E}_{p_{0t}}\left[\left|b_i(x_0 - t\mathrm{grad}f(x_0) + \sqrt{2t}z)\right|\right] \leq tK_4 \tag{53}$$

where $K_4$ is determined by the expectation of $\langle\nabla b_i(x_0), \nabla f(x_0)\rangle$ and $\langle z, \nabla^2 b_i(x_0)z\rangle$. So we have

$$\int_{\mathbb{R}^n}\left(\int_{\mathbb{R}^n}\left(\sum_{ij} b_i \frac{\partial p_{0t}(x_0,x)}{\partial x_i}\right)\sqrt{|g(x_0)|}dx_0\right)\log\frac{p_t}{\nu}\sqrt{|g(x)|}dx \leq tnK_3K_4.$$

$$\frac{d}{dt}H(p_t|\nu) = \int_{\mathbb{R}^n}\left(\int_{\mathbb{R}^n}\left(\mathrm{div}(p_{t|0}(x|x_0)\mathrm{grad}f(x_0)) + \Delta_M p_{t|0}(x|x_0)\right)p_0(x_0)\sqrt{|g|}dx_0\right)\log\frac{p_t}{\nu}dx \tag{54}$$

$$\frac{d}{dt}H(p_t|\nu) \leq -\frac{3}{4}J + \frac{4t^2 L^4}{\alpha}H(p_0|\nu) + 2t^2 nL^3 + 2tnL^2 \tag{55}$$

$$+ n^2 K_2 \mathbb{E}_{p_{t0}}\left[O\left(\left\|-t\mathrm{grad}f(x_0) + \sqrt{2t}z\right\|^2\right)\right] + tnK_3K_4 \tag{56}$$

$$\leq -\frac{3}{4}J + \frac{4t^2 L^4}{\alpha}H(p_0|\nu) + 2t^2 nL^3 + 2tnL^2 \tag{57}$$

$$+ n^2 K_2 C\left(\frac{4t^2 L^2}{\alpha}H(p_0|\nu) + 2t^2 nL + 2tn\right) + tnK_3K_4 \tag{58}$$

$$= -\frac{3}{4}J + \frac{4t^2 L^4 + 4t^2 L^2 n^2 K_2 C}{\alpha}H(p_0|\nu) + 2t^2(nL^3 + n^3 K_2 CL) + t(2nL^2 + 2n^3 K_2 C + nK_3K_4) \tag{59}$$

Let $t \leq \epsilon \leq \frac{2nL^2 + 2n^3 K_2 C + nK_3K_4}{2(nL^3 + n^3 K_2 CL)}$, we have

$$\frac{d}{dt}H(p_t|\nu) \leq -\frac{3\alpha}{2}H(p_t|\nu) + \frac{4\epsilon^2(L^4 + L^2 n^2 k_2 C)}{\alpha}H(p_0|\nu) + 2\epsilon(2nL^2 + 2n^3 K_2 C + nK_3K_4) \tag{60}$$

Multiplying both sides by $e^{\frac{3\alpha}{2}t}$, we have

$$\frac{d}{dt}\left(e^{\frac{3\alpha}{2}t}H(p_t|\nu)\right) \le e^{\frac{3\alpha}{2}t}\left(\frac{4\epsilon^2(L^4+L^2n^2k_2C)}{\alpha}H(p_0|\nu)+2\epsilon(2nL^2+2n^3K_2C+nK_3K_4)\right)$$

and integrating for $t \in [0,\epsilon]$,

$$e^{\frac{3}{2}\alpha\epsilon}H(p_\epsilon|\nu)-H(p_0|\nu) \le \frac{2(e^{\frac{3\alpha}{2}\epsilon}-1)}{3\alpha}\left(\frac{4\epsilon^2(L^4+L^2n^2k_2C)}{\alpha}H(p_0|\nu)+2\epsilon(2nL^2+2n^3K_2C+nK_3K_4)\right) \tag{61}$$

$$\le 2\epsilon\left(\frac{4\epsilon^2(L^4+L^2n^2k_2C)}{\alpha}H(p_0|\nu)+2\epsilon(2nL^2+2n^3K_2C+nK_3K_4)\right) \tag{62}$$

So

$$H(p_\epsilon|\nu) \le e^{-\frac{3}{2}\alpha\epsilon}\left(\frac{8\epsilon^3(L^4+L^2n^2K_2C)}{\alpha}+1\right)H(p_0|\nu)+e^{-\frac{3}{2}\alpha\epsilon}4\epsilon^2(2nL^2+2n^3K_2C+nK_3K_4) \tag{63}$$

$$\le e^{-\frac{3}{2}\alpha\epsilon}\left(\frac{8\epsilon^3(L^4+L^2n^2K_2C)}{\alpha}+1\right)H(p_0|\nu)+4\epsilon^2(2nL^2+2n^3K_2C+nK_3K_4). \tag{64}$$

If $1+\frac{8\epsilon^3(L^4+L^2n^2K_2C)}{\alpha} \le 1+\frac{\alpha\epsilon}{2} \le e^{\frac{1}{2}\alpha\epsilon}$, or $\epsilon \le \frac{\alpha}{4L\sqrt{L^2+n^2K_2C}}$,

$$H(p_\epsilon|\nu) \le e^{-\alpha\epsilon}H(p_0|\nu)+4\epsilon^2(2nL^2+2n^3K_2C+nK_3K_4),$$

and then

$$H(p_{k+1}|\nu) \le e^{-\alpha\epsilon}H(p_k|\nu)+4\epsilon^2(2nL^2+2n^3K_2C+nK_3K_4).$$

$\square$

**Theorem 4.3.** Suppose $M$ is a compact manifold without boundary and $R$ is the Riemann curvature, $\nu = e^{-f}$ a density on $M$ with $\alpha > 0$ the log-Sobolev constant. Then there exists a global constant $K_2, K_3, K_4, C$, such that for any $x_0 \sim \rho_0$ with $H(\rho_0|\nu) \le \infty$, the iterates $x_k \sim \rho_k$ of GLA with stepsize $\epsilon \le \min\{\frac{\alpha}{4L\sqrt{L^2+n^2K_2C}}, \frac{2nL^2+2n^3K_2C+nK_3K_4}{2(nL^3+n^3K_2CL)}, \frac{1}{2L}, \frac{1}{2\alpha}\}$ satisfty

$$H(p_k|\nu) \le e^{-\alpha k\epsilon}H(p_0|\nu)+\frac{16\epsilon}{3\alpha}(2nL^2+2n^3K_2C+nK_3K_4)$$

*Proof.*

$$H(p_k|\nu) \le e^{-\alpha k\epsilon}H(p_0|\nu)+\frac{1-e^{-\alpha k\epsilon}}{1-e^{-\alpha\epsilon}}4\epsilon(2nL^2+2n^3K_2C+nK_3K_4) \tag{65}$$

$$\le e^{-\alpha k\epsilon}H(p_0|\nu)+\frac{4}{3\alpha\epsilon}4\epsilon^2(2nL^2+2n^3K_2C+nK_3K_4) \tag{66}$$

$$= e^{-\alpha k\epsilon}H(p_0|\nu)+\frac{16\epsilon}{3\alpha}(2nL^2+2n^3K_2C+nK_3K_4) \tag{67}$$

$\square$

# D Experiments

As mentioned before, for simplicity, we can implement GLA without using the exponential map where a geodesic ODE solver is required, especially for the case when $M$ is a submanifold of $\mathbb{R}^n$. In general, the retraction map from $T_x M$ to $M$ is used in optimization on Riemannian manifold [**?** ], as a replacement of exponential map. In this section, we give experiments on sampling from distributions on the unit sphere in comparison of exponential map and orthogonal projection as a retraction in the geodesic step of GLA.

The experiments are designed to verify the following properties:

1. GLA captures the target distribution $e^{-f}$ as expected;
2. The projection map behaves well in replacing the exponential map without solving geodesic equations.

In each set of figures, (a) is the landscape of the ideal distribution, (b) and (c) are the results with small number of iterations for exponential map and projection, (d) and (e) are enhanced with large number of iterations. We start with the definition of the general retraction in optimization on manifold.

**Definition D.1** (Retraction). A retraction on a manifold $M$ is a smooth mapping $\mathrm{Retr}$ from the tangent bundle $TM$ to $M$ satisfying properties 1 and 2 below: Let $\mathrm{Retr}_x : T_x M \to M$ denote the restriction of $Retr$ to $T_x M$.

1. $\mathrm{Retr}_x(0) = x$, where $0$ is the zero vector in $T_x M$.

2. The differential of $\mathrm{Retr}_x$ at $0$ is the identity map.

Suppose $M$ is a submanifold of $\mathbb{R}^n$ with positive codimension. Denote $\mathrm{Proj}_{T_x M}$ the orthogonal projection to the tangent space at $x$, then the retraction can be defined as $\mathrm{Retr}_x(v) = \mathrm{Proj}_M(x + v)$. The GLA on a submanifold of $\mathbb{R}^n$ can be written as

$$x_{k+1} = \mathrm{Retr}_x \left( \mathrm{Proj}_{T_x M}(-\epsilon \nabla f(x_k) + \sqrt{2\epsilon}\xi_0) \right) \tag{68}$$

If $M = S^{n-1}$ be the unit sphere in $\mathbb{R}^n$, then $\mathrm{Retr}_x(v) = \frac{x+v}{\|x+v\|}$.

(a) Ideal distribution of $e^{-f}$

(b) $\mathrm{Exp}_x(v)$, iteration: 10k

(c) $\mathrm{Retr}_x(v)$, iteration: 10k

(d) $\mathrm{Exp}_x(v)$, iteration: 100k

(e) $\mathrm{Retr}_x(v)$, iteration: 100k

Figure 1: $f(x) = x_1 + x_2 + x_3$, stepsize $\epsilon = 0.1$.

(a) Ideal distribution of $e^{-f}$

(b) $\mathrm{Exp}_x(v)$, iteration: 10k

(c) $\mathrm{Retr}_x(v)$, iteration: 10k

(d) $\mathrm{Exp}_x(v)$, iteration: 100k

(e) $\mathrm{Retr}_x(v)$, iteration: 100k

Figure 2: $f(x) = x_1^2 + 3.05x_2^2 - 0.9x_3^2 + 1.1x_1x_2 + -1.02x_2x_3 + 2.1x_3x_1$, setpsize $\epsilon = 0.1$.