[Reviews · NeurIPS 2020]

Review 1

Summary and Contributions: The paper is interesting, presenting a geometric convergence result for discrete diffusions on manifolds (geodesic Langevin algorithm, essentially the most natural extension of Euler Maruyama to a manifold). The results presented are, however, more theoretical than practical and the experiments, presented only in supplement, may be too trivial to be useful in assessing the practical importance for typical NIPS readership.

Strengths: The results are well presented and all mathematical details appear to be correct. Theorem 4.2 is useful basic result which should be of widespread interest. The topic is important: constraints are increasingly relevant in machine learning and statistical sampling applications. The article therefore offers an important step in the understanding of a basic algorithm of fundamdental importance.

Weaknesses: The bound on stepsize in Theorem 4.4 relies on several estimates and is unlikely to be tight in practice (or calculable). The numerical experiment presented in supplement, for a projected Euler Maruyama method, is only for sampling an artificially constructed distribution on a sphere. The demonstration of convergence rate is not shown. Momenta (underdamped formulation of Langevin dynamics), which is often superior for sampling, is not considered.

Correctness: Results appear to be correct, although I did not check all details.

Clarity: yes, and well presented for a broad audience.

Relation to Prior Work: There does not seem to be a rigorous treatment of the LSI for the GLA in the literature. Basic existing results on diffusions on manifolds seem to be discussed. There are a number of related methods and results in the literature for specific cases (e.g. Piggot and Solo SIAM Num. Anal 2016, gives related results for specific manifolds), but no previous paper seems to give the sort of analysis presented here.

Reproducibility: Yes

Additional Feedback: I think the primary weakness of the article is lack of complex numerical study which convinces machine learning audiences of the practical relevance of the estimates obtained.


Review 2

Summary and Contributions: In this paper, the authors propose generalizes the results of the existing work by Vempala and Wibisono 2019 to sampling from distributions on manifolds. They give a non-asymptotic convergence guarantee for this with the help of log-Sobolev inquality.

Strengths: The proposed method allows as to perform sampling from distributions on manifolds without solving ODE. Although the proposed sampling is biased, its error degree can be set up. With the help of log-Sobole inequality, they show non-asymptotic convergence guarantee for GLA, which could be useful to understand the rate of convergence in sampling from distributions on manifolds and could be the theoretical bases for further study on this topic. Including these, the paper provides theoretical principles necessary for their sampling scheme.

Weaknesses: The paper provides no experimental result (although some in the supplementary) supporting the validity of their sampling method. It would be appreciated to provide it for more understanding of the proposed one (since, in my understanding, the implementation would not necessarily difficult and thus it also can be an advantage of their method).

Correctness: Although I cannot understand the details of the mathematical contexts in the proofs, the mathematical descriptions look solid.

Clarity: I think the readability is well enough for its mathematical contexts. However, I think it is not clear how each mathematical content contribute to the practical motivation with the current description of this paper.

Relation to Prior Work: The paper describes clearly the relation with existing works and their contributions upon those.

Reproducibility: Yes

Additional Feedback:


Review 3

Summary and Contributions: The authors propose a method to sample from an exponential distribution that is defined over a Riemannian manifold. The sampling scheme is based on the Langevin dynamics and utilizes the geodesics on the manifold. Additionally, convergence guarantees are provided. ===== After rebuttal ===== I would like to thank the authors for their responses. My main concerns are generally about the clarity and the structure of the paper. I believe that the submission needs several improvements to that extend. Even after the author responses I am not sure how this can be addressed in the camera-ready version. Therefore I retain my score.

Strengths: The problem that the authors aim to solve is interesting. Also, the technical part of the paper seems to be solid.

Weaknesses: I think that the paper is very hard to read for non-experts, since there are a lot of technical details. The mathematical content is not so easily accessible.

Correctness: The technical part of the submission seems to be correct. However, I find it very hard to verify the technical part of the paper, especially in such a short time.

Clarity: I think that the paper is very hard to read for the average machine learning researcher. There is a lot of compressed mathematical content, which is not easily accessible. In my opinion, the manuscript can be improved such that to be more easy to read from a general audience.

Relation to Prior Work: All the related work seems to be included. However, it is not clear in my opinion which are the differences.

Reproducibility: No

Additional Feedback: - One of the problems I had while reading the paper is to understand the notation, I think it is quite abstract. - As a non-specialized reader, I think that it is not clear from the text in which space each quantity is defined e.g. geodesics, tangent vectors, grad, dX_t, etc. For instance, the geodesic in line 48 seems to be in local coordinates, while in other parts seems to be computed directly on the manifold. - I would be really happy to know what is the algorithm that you propose. Especially, if this can be described explicitly e.g. what is the dimensionality of the gradients, what is the noise vector, what is dimension of the tangent vector, etc. I am pretty sure that the technical content is correct. Also, I think that the convergence results are interesting. However, it is particularly hard to read and to understand what actually happens in the paper. Probably one reason is that I am not an expert in this field. However, I believe that the content of the paper should be accessible such that to be an impactful contribution.


Review 4

Summary and Contributions: This paper deals with the problem of sampling from a probability distribution over a Riemannian manifold, mainly carrying over to this case the results that Vampala and Wibisono (2019) developed for distributions over R^n. The considered distribution has unnormalized density e^{-f(x)} and is assumed to satisfy a Logarithmic-Sobolev inequality (LSI), while f is not assumed to be convex, which means that the target distribution is not assumed to be log-concave. The finite-time approximations are based on the unadjusted Langevin dynamics (ULA). The main results, like Vempala and Wibisono (2019), establish convergence in KL divergence of the finite-time approximations to the target distribution.

Strengths: Interesting mix of mathematical areas and concepts. The mathematical derivations look believable (I did not check every detail, but I am familiar with Vempala and Wibisono's work and the arguments follow similar steps).

Weaknesses: The relevance of carrying over the sampling problem to a Riemannian manifold is not motivated/discussed, nor illustrated, hence the connection with machine learning is weak. The results do not add to what was established by Vempala and Wibisono (2019).

Correctness: The math looks correct.

Clarity: Unfortunately not.

Relation to Prior Work: Reasonably well.

Reproducibility: Yes

Additional Feedback: The novel work was done by Vempala and Wibisono (2019) for distributions on R^n, while this paper carries over the results from R^n to an n-dinemsional Riemannian manifold. Specifically, Theorem 4.2 carries over Vempala and Wibisono (2019)'s Theorem 1, and Theorem 4.3 carries over Vempala and Wibisono (2019)'s Theorem 2, to a Riemannian manifold. However, the results of this paper do not add much to what Vempala and Wibisono (2019) had established. Of course seeing the results carried over to a Riemannian manifold is mathematically interesting, but on the other hand this is straightforward. Lacking a clear motivation for this extension to a Riemannian manifold, what I see is an interesting mathematical curiosity. To make this paper relevant for the NeurIPS readership I suggest illustrations of specific problems where sampling from a Riemannian manifold occurs naturally, or perhaps some discussion how carrying out the analysis on a manifold brings something new (result or insight). On this note, a relevant but missing reference is Ikeda and Watanabe's book "Stochastic differential equations and diffusion processes" for the material on SDE's on Riemannian manifolds. I also suggest to correct/clarify at several places this: e^{-f} is the unnormalized density, hence the measure with density e^{-f} is not a probability distribution. Note that also writing "\nu = e^{-f}" is misleading, the measure associated to the density e^{-f} is defined by integrals: \nu(A) = \int_{A} e^{-f(x)} dx, with "dx" for the natural volume measure on your space. The corresponding probability measure would be obtained upon normalizing by \nu(M), provided e^{-f} has a finite integral. I also suggest refactoring the writing/presentation of the paper to make it appealing to a machine learning readership. I did not understand how the paper is connected to this: ". With the development of continuous game theory and GANs in recent years, our results have potential impact to solve the Nash equilibrium of the games with continuous strategy space." *** AFTER AUTHOR RESPONSE *** Thank you for taking the time to respond to my comments. From the discussions it is apparent that this paper may have contributions of potentially sufficient value. However, the main concern remains that the submitted paper was poorly written, and it needs a considerable amount of changes to reach a satisfying version. Unfortunately is not clear if this can be achieved in a single round, which is why I continue to argue for rejection.

[Author Response · NeurIPS 2020]

We thank the reviewers for their work, especially these difficult times. Our paper uses mathematical tools that many
researchers in ML/AI might not be very familiar with. We have added some background both in the main part and in
the supplementary material (due to space restrictions, it is impossible to put all definitions in the main part).

**[To Reviewer #1]** We thank you for your careful reading, supportive comments, and clarifying the relation to prior
works. Especially we appreciate your suggestion of underdamped Langevin for further consideration.

**[To Reviewer #3]** We are considering experiments based on stereographic projection of sphere to illustrate the
equivalence of the implementations of our algorithm. Thanks for pointing this out.

**[To Reviewer #4]** Dimension of gradient and tangent vectors are equal to the dimension of the manifold. The noise
vector is the Euler-Maruyama discretization of Riemannian Brownian motion.

**[To Reviewer #5]** The main points raised are:
1) Sampling on Riemannian manifold lacks of motivation and connection with ML;
2) The results of our paper carry over those of Vempala and Wibisono(2019) to manifold, and obtaining these results on
manifold is straightforward.

Regarding the first point, we name a few papers to clarify the ecology of sampling on Riemannian manifold. Girolami
and Calderhead[2011] "Riemannian manifold langevin and hamiltonian monte carlo methods", has 1200 citations,
amongst, 30 from NIPS, 17 from ICML, 8 from JMLR, 9 from AISTATS, 5 from AAAI and 1 from COLT. Byrne
and Girolami[2013]"Geodesic monte carlo on embedded manifold", 107 citations, 4 from NIPS,4 from AAAI, 3 from
JMLR and 2 from ICML. Data from a rough counting through google scholar.
Brubaker et al [2012]"A family of MCMC methods on implicitly defined manifolds", AISTATS, 65 citations.
Patterson and Teh "Stochastic gradient Riemannian langevin dynamics on the prob. simplex", NIPS13, 198 citations.
Lan et al [2014]"Spherical Hamiltonian monte carlo for constrained target distributions", ICML, 40 citations.
Liu et al "Stochastic gradient geodesic MCMC methods", NIPS16, 19 citations.
Smith et al "Stochastic natural gradient descent draws posterior samples in function space", NIPS18.
Goyal, Shetty "Sampling and optimization ... in Riemannian manifolds of nonnegative curvature", COLT19.
Zhang et al "Wasserstein control of mirror langevin monte carlo", COLT20.
All the papers are about sampling on Riemannian manifold. It is a well established and active research area of ML.
"I suggest illustrations...occurs naturally," Sampling as a generic technique of generating random points has many
applications. Sampling on manifold plays a role whenever constraints are introduced, according to the above papers,
these constraints include: simplex with Fisher information metric, sphere, truncated set of $\mathbb{R}^d$ with Hessian metric,
general embedded manifold in $\mathbb{R}^d$, etc. They have variety of application backgrounds: posterior sampling, molecule
dynamics, truncated statistics, and an important example of embedded manifold is the matrix manifold with applications
in robotics and computer vision. We motivate in our paper at line 21-27 with PCA, matrix completion, matrix
factorization based on the work of Moitra and Risteski [2020], and continuous game theory based on the work of
Domingo-Enrich et al[2020] where manifold langevin is used in zero-sum game, while our result is the potential game
counterpart. "I did not understand...", they are connected since manifold Langevin descent-ascend is used in their paper
to sample from stationary distribution, and our technique can give insight to the rate of convergence for their method.

About the second point, we argue that the analysis to generalize the result from Euclidean to Riemannian manifold is
precisely the contribution, since to our best knowledge, this type of result does not exist in the aforementioned papers.
To support the claim that this is not straightforward, we have to point out that the result and the approach of Vempala
and Wibisono for $\mathbb{R}^d$ do not hold for general manifold. In fact, it is a very open ended problem for one to find reasonable
conditions under which certain convergence results can be proved, and any success of doing this can be considered
significant to some extent. For example, the aforementioned paper Goyal and Shetty, COLT 2019 and the following

Mangoubi and Smith [2018]"Rapid mixing of geodesic walk on manifold with positive curvature", Annals of Applied
Probability, address the uniform distribution on manifold(and its subsets) with positive/non-negative curvature condition.
Orthogonal to their approach, we put topological constraints(line 242: Assumption 1) instead of curvature constraints.
Furthermore, the main technique in Vempala and Wibisono's is to control the error caused by discretization, while in
our paper, the main technique is to control the error caused by linearization of a curved space, and it needs a careful
treatment with very different toolkits(geodesic equations, sectional curvature, Riemann curvature, Jacobi field, parallel
transport, etc) compared to Euclidean space. For example, the proof of Theorem 4.4 is based on Jacobi field method
that has no trivial connection to Vempala and Wibisono's technique.

Again we emphasize that one contribution of our paper is to provide a general framework with necessary techniques that
will be useful in other contexts as well, e.g. adjusted Langevin and Hamiltonian Monte Carlo with manifold constraints.
This is also part of the role of a theory paper: to find techniques that might be leveraged by others in different contexts.
The list of papers shows that ML community has a growing interest in manifold sampling but the convergence results
are much fewer than that for Euclidean space. Filling up the technical gap is exactly the motivation and contribution.

[Meta-Review · NeurIPS 2020]

This paper extends/generalizes the results from Vempala and Wibisono (2019) on Langevin diffusions to Riemannian manifolds. The problem is quite interesting, has many potential applications and the derivations are sound. The result relating the diameter of a compact manifold to the Langevin step-size is neat. There is overall concern between reviewers that accessibility of the current draft to the broader Neurips community is rather limited.